# Aspects Regarding Sustainability among Private Dental Practitioners from Bucharest, Romania: A Pilot Study

**DOI:** 10.3390/healthcare11091326

**Published:** 2023-05-05

**Authors:** Ana Maria Cristina Țâncu, Andreea Cristiana Didilescu, Mihaela Pantea, Ruxandra Sfeatcu, Marina Imre

**Affiliations:** 1Department of Prosthodontics, Faculty of Dentistry, “Carol Davila” University of Medicine and Pharmacy, 17-23 Calea Plevnei Street, Sector 1, 010221 Bucharest, Romania; anamaria.tancu@umfcd.ro (A.M.C.Ț.); marina.imre@umfcd.ro (M.I.); 2Department of Embryology, Faculty of Dentistry, “Carol Davila” University of Medicine and Pharmacy, 17-23 Calea Plevnei Street, Sector 1, 010221 Bucharest, Romania; 3Department of Oral Health and Community Dentistry, Faculty of Dentistry, “Carol Davila” University of Medicine and Pharmacy, 17-23 Calea Plevnei Street, Sector 1, 010221 Bucharest, Romania; ruxandra.sfeatcu@umfcd.ro

**Keywords:** dentistry, sustainability, hazardous waste, COVID-19, professional role, Romania

## Abstract

Oral health professionals’ knowledge of sustainability is essential for promoting environmental protection in dental healthcare. This pilot study involved an online survey addressed to 70 dental private practitioners from Bucharest, Romania, to evaluate their awareness of the concept of sustainability in dentistry. The performed statistical analysis revealed that 41.4% of the participants were well aware of sustainability in dentistry, with older participants demonstrating significantly higher levels of such awareness (*p* = 0.001). Sustainability awareness among participants correlates positively with their knowledge of the negative environmental impacts of dental activity (*p* < 0.001) and with the concern for sustainable dentistry implementation in their workplace (*p* = 0.037). Improper biohazardous waste disposal was identified as the primary cause of negative environmental impact of dental practices by 87.1% of participants. Installing high energy-efficient dental equipment was selected as the most important action to implement sustainability in participants’ dental practices (64.3%). Overall, 51.4% of the participants reported that the COVID-19 pandemic had a medium impact on their dental activity in terms of sustainability. Our study found that participants have a moderate level of awareness regarding sustainability in dentistry, highlighting the need for education on sustainability for oral health professionals.

## 1. Introduction

Oral healthcare professionals are in a position to provide dental services that are environmentally sustainable. Dentists have an ethical responsibility to ensure that dental activities are done in a sustainable manner, in accord with the third pillar of the FDI’s (World Dental Federation) vision for 2030, which acknowledges the urgent need for environmentally friendly and people-centered dental care [1,2].

Sustainability concerns date back to the early 1990s, starting with the United Nation (UN) Agenda for the 21st century and continuing with the 17 Sustainable Development Goals (SDGs) to be reached by the year 2030 [3]. There is a need to care for the environment, to preserve the earth’s resources and to sustain knowledge and practices for global sustainability [4,5]. Education is the key for supporting sustainable development under the aegis of the education for sustainable development (ESD) concept that promotes principles, values, knowledge and skills in relation to the sustainability behavior of students and health professionals [3,6,7,8]. Research, teaching and learning are sources of knowledge and methods of changing behavior to provide action for sustainable development [3,9,10,11]. In daily dental practice, there are some barriers but also some opportunities and solutions for sustainability in dentistry in order to preserve life and health, and support the green economy. In the field of dentistry, there has been a continuous focus on providing the best possible patient care, regardless of the impact on the environment. It is essential to raise awareness among dental professionals about the key parameters necessary for effective and sustainable practices, including reducing the amount of carbon dioxide equivalents released into the air and water, as well as improving waste management strategies [4,12].

It is important to reconsider the concept of sustainability in dentistry, particularly in the context of the modernization of dental practices which involve the use of single-use and disposable items. The COVID-19 pandemic has further complicated this issue, as legislation has recommended to increase the usage of single-use plastics (SUPs) and protective equipment (PPE); these practices have contributed to the environmental footprint of dental activities [12]. It is important to manage the production but also the disposal of PPE products in order to reduce environmental pollution [13,14,15]. There is a need to reduce plastic use and consumption by investing in renewable alternative materials, but it is important as well to improve its management (by disposing and degrading together with the organic waste rather than by incineration) [13].

There are three main sources for CO_2_ emissions in dentistry: patients and dental team transit to and from dental practices, supply chain for dental materials, and management waste [1]. In order to provide high-quality dental care, there must be a focus on four areas of activity, implying preventive, operative, integrated care, and ownership of care with two environmental outcomes: reduced CO_2_ emissions (fewer patients appointment, less manufacturing and resources distributions) and decreased waste and pollution (less materials, packaging and clinical waste) [1,4].

While aiming to support sustainability and decrease the environmental impact at the international level, two important and recognized policies have been developed and established since 2015, namely, the Paris Agreement and the United Nations 2030 Agenda for Sustainable Development [16,17]. Other community-based initiatives are also effective due to the fact that they provide solutions that are appropriate to the local context [3,18]. Recently, the COVID-19 pandemic has increased the importance of sustainability concepts, mainly the environmental impact from a healthcare point of view [15]. Regarding the dentistry field, there is a need for change in order to decrease the environmental footprint. University institutions embrace the importance of sustainable workplaces for teaching staff and dental students, along with including sustainability concepts in curriculum [11,17]. The Association for Dental Education in Europe (ADEE) has an important theme of discussions regarding courses on teaching sustainability in dentistry [12]. To address the environmental impact of dental practices, there is a growing interest in including four major topics into undergraduate dental curricula: oral health promotion and prevention (minimal intervention principles; targeted individualized treatment plans); patient education (empowerment for daily oral care at home); lean service delivery (optimization of dental appointments in order to reduce patient travel; reducing waste by management of stocks); preferential use of strategies with decreased impact on the environment (digital dentistry, tele-dentistry, and avoiding the use of amalgams) [11,12,15,17].

These topics on sustainability are of interest for the dental professionals, but also for the patients who need to be educated about preventive practices, attitudes and behaviors related to preserving optimal oral health throughout life [10,14]. Education represents the key concept for raising awareness for dental professionals, undergraduate students and dentists, of sustainable dental practices through degree programs and continuing medical education [14]. Likewise, patients’ empowerment for an active role in the management of their own state of health is an extremely important aspect [14]. Dental practitioners need to be aware of updated legislation related to sustainability in order to ensure a safe, evidence-based and sustainable dental practice. The dental team needs to be able to think critically about the impact of clinical procedures and laboratory work on the environment and to achieve sustainable results reflected in decreased waste and pollution, and reduction of CO_2_ emissions [1,4]. Dentists are responsible for applying sustainability in their settings, and the educators must develop appropriate teaching materials to be included in educational dental curricula; nevertheless, scientific research on sustainability needs to be prioritized [11,17]. Putting sustainability behaviors into practice is challenging, especially in developing countries, given several recognized barriers: lack of a knowledge base, economic issues and legislative initiatives [12,19]. To our knowledge, dental scientific literature provides relatively few scientific studies dedicated to issues concerning sustainability in dentistry in Romania and the degree of awareness regarding the concept of sustainability among Romanian dentists.

Giving this context, the present research aims to evaluate the level of awareness regarding the concept of sustainability in dentistry within a sample of dentists from Bucharest, Romania. Moreover, this study investigates these dentists’ opinions regarding the negative impact of dental activities on the environment as well as the methods they prefer to use in implementing sustainable practices within their profession.

## 2. Materials and Methods

### 2.1. Survey Methodology

The study was conducted in the Faculty of Dental Medicine, “Carol Davila” University of Medicine and Pharmacy, Bucharest, Romania. This survey was approved by the Scientific Research Ethics Committee of “Carol Davila” University of Medicine and Pharmacy, Bucharest, Romania (Protocol number: 23829/20.09.2021). The study was conducted in accordance with the Declaration of Helsinki of 1975, revised in 2013. Subjects selected to participate in the study were invited to fill in the questionnaire and were informed about the survey in respect to the World Medical Association Declaration of Helsinki and the current European privacy law. In this survey, we aimed to evaluate the level of awareness regarding sustainability among a sample of dentists located in Bucharest, the capital city of Romania. The research had a cross-sectional design. The survey was designed as a pilot study, and it is planned to be expanded to include other regions within Romania for a more comprehensive analysis. The questionnaire was created in Romanian and was constructed using Google Forms (Mountain View, CA, USA, Alphabet). The formulation of the questionnaire was conducted with care to exclude any instances of multiple negations, misleading questions, or vague terminologies. The questionnaire underwent a face validity review by two experts in the field of dental medical education. The invitation to participate in the survey, written information about the study, informed consent, and link to the self-administered Google Forms questionnaire were distributed via email. The survey was sent out during the first two weeks of October 2021. All subjects that agreed to participate in the study expressed their consent by completing the survey. No personal data were collected through the form and, as an anonymous web survey, no sensitive data were collected.

### 2.2. Survey Population

An active recruitment of subjects was carried out at the initiative and under the direction of the investigators of the study. The inclusion criteria employed for participant selection in this study were as follows: dentists who are Romanian citizens and residing in Bucharest, aged between 18 and 65 years, and have graduated from the Faculty of Dentistry at “Carol Davila” University of Medicine and Pharmacy in Bucharest, Romania. Only former or current residents in dentistry at the previously mentioned university and active dental private practitioners were included in this study. The exclusion criteria for this study comprised individuals who declined to provide their informed consent to participate, as well as those who exhibited non-compliance with the study’s investigative teams. The request to participate in this survey was applied to 85 dentists, of which 70 have voluntarily agreed to participate, which means an acceptance of participation of 82.35%. All respondents delivered answers to all questions in the questionnaire and no data were eliminated.

### 2.3. Survey Questionnaire

The questionnaire used for the assessment was formed of 10 items represented by single- and multiple-choice questions, referring to the following main aspects (Table 1):(1)sociodemographic data referring to age, gender and professional experience measured in years (3 questions);(2)dentists’ general level of awareness regarding the concept of sustainability in dentistry (2 questions);(3)dentists’ opinion regarding the negative impact of their dental activities on the environment (2 questions);(4)dentists’ opinion regarding the most effective approaches for implementing sustainability into dental practices (2 questions);(5)dentists’ opinion on the COVID-19 pandemic impact on sustainability in dentistry (one question).

Transportation and frequency of visits to the dental practice are acknowledged as significant sustainability issues in dentistry; however, the survey’s questions were intentionally designed to focus on particular issues that are strongly related to the routine dental procedures carried out by the participating practitioners and that are entirely dependent on their decisions and actions. Therefore, we do not include questions about travel-related issues in our questionnaire.

Items 1–4 and 7 were represented by single-choice questions. The other items were represented by multiple-choice questions. The participants were asked to select, for questions 8 and 9, a maximum of three answers from the provided options. The estimated completion time for the questionnaire was 10 min.

### 2.4. Data Analysis

All the data from the study were analyzed using IBM SPSS Statistics 25. Quantitative variables were tested for normal distribution using the Shapiro–Wilk Test and were written as averages with standard deviations or medians with interquartile ranges. Qualitative variables were written as counts or percentages. Quantitative independent variables were tested using Mann–Whitney U/Kruskal–Wallis H Tests according to their non-parametric distribution. Qualitative variables were tested using Fisher’s exact tests/Pearson chi-square tests.

## 3. Results

The performed statistical analysis revealed the following presented elements: The participants average age was 32.43 ± 7.76, (26.7–35.2), ranging from 24 to 53 and with a median of 30 years. Most of the analyzed participants were women (82.9%, n = 58) and, regarding the professional experience, the average duration of their period of professional activity in the field of dentistry was 7.03 ± 7.61 years with a median of 4 years, at the moment the survey was administered.

Data from Table 1 show the distribution of the participants based on the answers collected from the survey. The results show the following:-Only 41.4% of the dentists considered that they are well aware of the concept of sustainability in dental practice; the most frequent answers related to the aspects that represent sustainability in today’s dental practice were: reduction of the impact of dental activity on the environment (82.9%) and reduction in the amount of wasted energy, electricity, water and paper (62.9%). All possible answers that were assigned to these question were valid/correct, therefore, a score for measuring participants’ general level of awareness regarding sustainability was constructed (SCORE_ED) based on how many answers were selected. The average score was 1.96 ± 1.05 with a median of 2 (41.4% had 1 option selected, 37.1% had 2 options selected, so 78.5% of the dentists had 2 options or fewer selected). This means that, in this study, the general level of awareness regarding sustainability is moderate (based on half of the answers selected in general);-Participants’ most frequent answers regarding the negative impact of their dental activities on the environment were as follows: improper disposal of biohazardous waste (87.1%); energy, electricity, water and paper waste (67.1%); and improper use and recycling of dental amalgam (58.6%). All possible answers that were assigned to these questions were valid/correct, therefore, a score for measuring participants’ general level of awareness regarding the negative impact of dental activity on the environment was constructed (SCORE_IM) based on how many answers were selected. The average score was 3.41 ± 1.82 with a median of 3 (52.9% of the dentists had 3 options or fewer selected). This means that, in this study, the general level of awareness regarding the negative impact is moderate (based on half of the answers selected in general);-Overall, 61.4% of the dentists considered that, at their workplace, there are general concerns for sustainable dentistry;-The most frequent answers regarding the most effective approaches for implementing sustainability into dental practices were as follows: reduction of paper usage (72.9%), reduction of energy/electricity consumption (62.9%) and the use of digital equipment and technologies (60%);-The most frequent reported actions that participants intend to take so as to implement sustainability in their dental practice include the following: the installation of dental equipment with high energy efficiency (64.3%), the implementation of practices related to reusing and recycling (55.7%), the adoption of strategies focused on energy and water resource conservation (52.9%), as well as the use of environmentally friendly or “green” products (52.9%);-Overall, 51.4% of the dentists considered that the COVID-19 pandemic had a medium impact on their dental activity in terms of sustainability, and 45,7% of the dentists considered that the COVID-19 pandemic had a high impact on the sustainability of the dental field.

Data from Table 2 and Figure 1, Figure 2, Figure 3 and Figure 4 show certain correlations between the analyzed parameters. All the analyzed variables have a non-parametric distribution according to the Shapiro–Wilk test (*p* < 0.001). According to the results, all the correlations observed are statistically significant (*p* < 0.05):-A statistically significant positive correlation exists between the age of dentists and the duration of their period of professional activity (*p* < 0.001, R = 0.879);-A statistically significant positive and weak correlation exists between the age of dentists and their level of awareness regarding the negative environmental impact of dental activities (SCORE_IM) (*p* = 0.001, R = 0.392). Although the correlation is weak, the data suggests that, on average, older dentists possess a greater level of awareness about the topic of environmental impact resulting from dental activity;-A statistically significant positive and weak correlation (SCORE_IM) (*p* = 0.006, R = 0.328) exists between the duration of professional activity and the level of awareness concerning the negative impact of dental procedures on the environment. Although the correlation is weak, the data suggests that, on average, dentists with longer periods of professional activity demonstrate a greater level of knowledge regarding the environmental negative impact of dental activity;-A statistically significant positive correlation (*p* < 0.001, R = 0.571) exists between participants’ general level of awareness concerning sustainability (SCORE_ED) and their level of awareness regarding the negative impact of dental activity on the environment (SCORE_IM). Specifically, the correlation is moderate in strength, suggesting that individuals with higher levels of awareness regarding sustainability are also more knowledgeable about the negative environmental impact of dental activity, and vice versa.

Data from Table 3 and Figure 5, Figure 6 and Figure 7 shows the comparison between the age of dentists, duration of their professional activity and certain answers collected in the survey. The analyzed variables have a non-parametric distribution in all groups according to the Shapiro–Wilk test (*p* < 0.05). According to the results, all the differences observed are statistically significant (*p* < 0.05):-Dentists who identified the use of traditional/conventional radiological systems or the use of disposable items during dental treatments as having a negative impact on the environment demonstrated a significantly higher age (median = 33 vs. 29; 32.5 vs. 29 years) (*p* < 0.001; *p* = 0.038) and a longer period of professional activity (median = 8 vs. 3; 7 vs. 3 years) (*p* = 0.003; *p* = 0.039) than those who did not share this perspective;-Dentists who selected the adoption of strategies focused on energy and water conservation as an approach to protect the environment had a significantly higher age (median = 32 vs. 29 years) (*p* = 0.048) and longer period of activity (median = 7 vs. 3 years) (*p* = 0.023) in comparison to those who did not share this viewpoint.

One important purpose of this study was to investigate potential statistically significant correlations between affirmative responses to question 4, which specifically addresses participants’ awareness concerning sustainability in dentistry, and their responses to other questions that may indicate the actual presence of such awareness. For example, data from Table 4 and Figure 8 indicate a relevant outcome: dentists who responded that there are concerns for sustainable dentistry implementation manifested at their workplace considered more frequently that they are well aware of the concept of sustainability in dental practice (75.9% vs. 51.2%) (*p* = 0.037).

## 4. Discussion

This paper aimed to identify awareness about the concept of sustainability among dental practitioners from the city of Bucharest, Romania. This pilot study was carried out during the first two weeks of October 2021, which was a period characterized by significant changes and unpredictability amid the COVID-19 pandemic. The responses collected from the applied survey indicate that the participants average age was 32.43 ± 7.76, the average duration of their period of professional activity in the field of dentistry was 7.03 ± 7.61 years at the moment when the survey was applied, and most of the participants were women (82.9%, n = 58). This survey gave us valuable feedback; hereunder, we present a synthesis of the main results of our pilot study correlated with relevant results of other studies collected from the scientific dental literature.

### 4.1. Dentists’ Awareness with the Concept of Sustainability in Dentistry

In our study, a percentage of 41.4 of the participants responded that they are well aware of the concept of sustainability in the field of dentistry. This outcome is relatively in line with the findings of similar studies reported in the scientific literature. For instance, a study conducted by Verma et al. (2020) [20], revealed that 52.5% of postgraduate participants and 48.4% of graduate participants were familiar with the term “green dentistry”. Similarly, a study conducted by Sen N. et al. [21] indicated that 60% of the participants (300 dentists, including dental practitioners and dental teaching staff) were aware of the term “green dentistry” [7,21]. On the other hand, the results of a recent study demonstrated that dental clinical directors from Portugal (n = 245) demonstrated good environmental awareness, 95.5% of the participants considered the implementation of environmental sustainability practices in the dental clinic as very important (38.8%) or important (56.7%) [22]. In regard to question 4 (“Are you well aware of the concept of sustainability in dentistry?”), we decided to use the term “well aware” in order to encourage participants to provide spontaneous and confident responses, while not concealing (minimizing) their potential knowledge limitations and to prevent any ambiguity or lack of rigor in the interpretation of their responses.

As a positive and consistent correlation, our results indicated that dentists who responded that there are concerns for sustainable dentistry implementation at their workplace considered more frequently that they are aware about the concept of sustainability in dentistry (*p* = 0.037). Moreover, the statistical score constructed for measuring the general level of awareness regarding sustainability in dentistry among participants (SCORE_ED) revealed a moderate level of awareness. In our interpretation, these results reflect an unfavorable situation. Sustainability awareness is not always highly reflected (“translated”) in the professional oral health domains [12]. As other authors have also noted [17,23,24,25,26], it is important that ideas related to sustainability in dentistry be integrated into both undergraduate dental curricula and postgraduate coursework in order to effectively incorporate the sustainable development goals into daily dental practice and facilitate the transition to a green economy, promoting healthy lifestyles and well-being throughout all life stages [12]. In the same line, Neves et al. [22] indicated that the main barriers to the implementation of environmental sustainability practices were the costs (44.6%) and the lack of information and training (16.3%).

Furthermore, the most frequent answers related to the aspects which represent sustainability in today’s dental practice were “reduction of the impact of dental activity on the environment” (58 answers/82.9% of 70 possible affirmative answers) and “reduction the amount of wasted energy, electricity, water and paper” (44 answers/62.9% of 70 possible affirmative answers). The findings of our study were consistent with those of other studies found in the scientific dental literature [12,27,28,29,30]; however, in our study, it was found that a minority of participants (21 answers/30% of 70 possible affirmative answers) agreed that the use of digital technologies has an influence on sustainability in routine dental practice. It is widely recognized that modern digital technologies have the potential to revolutionize dentistry, both in terms of education and clinical practice [31,32,33,34,35]. The digital transformation of dentistry encompasses advanced and strategic methodologies that have the capacity to deliver high-quality, efficient and sustainable dental care [32]. According to Lin et al. [36], various alternative approaches may be beneficial in improving the knowledge and practice of digital dentistry among dental practitioners; this area is currently undergoing a significant cultural shift with changes in how dental professionals, patients, service commissioners and policymakers approach digital transformation [31]. The need to demonstrate the safety, value, and utility of digital applications, to educate dental professionals and patients, to establish reliable and standardized processes and to encourage the adoption of open data and data sharing are just a few of the significant obstacles that are to be overcome [37]. Collaboration among academic institutions, industry, hospitals, non-profit organizations and government entities is essential in ensuring the effectiveness and affordability of digital health services [37,38].

Despite the fact that travel accounts for the biggest percentage of carbon emissions (carbon footprint) in dentistry (64.5%), as shown by a study commissioned by Public Health England [39], travel-related issues are not addressed in our questionnaire. The scientific literature highlights the fact that within dentistry, travel creates the highest carbon emissions and also contributes to human health damage [39]. Dental professionals can contribute to an improvement of environmental sustainability related to travel by combining family appointments, opting for alternative means of transportation, appropriate scheduling of dental appointments, using information technology and applying preventive dentistry [1,4,11]. In the present study, we opted to explore specific activities that occur within dental practices and are oriented towards treatment procedures. Our paper’s rationally narrowed focus allows us to explore sustainability aspects that are relatively underrepresented in the Romanian dental scientific literature to the best of our knowledge.

### 4.2. Dentists’ Opinions Regarding the Negative Impact of Dental Activities on the Environment

Dental practices generate substantial quantities of waste, encompassing a diverse range of categories such as recyclable materials, non-food domestic waste, hygiene waste, clinical, hazardous and food waste [40]. Proper management of waste is essential to avoiding pollution of the environment or harm to human health [40]. Effective waste management includes three main strategies: reducing waste generation; categorizing the waste and segregate the waste according to specific categories; and ensuring that the dental practices are evaluated by performing an audit (to assess the effectiveness of waste management practices in use) [40]. Based on the performed statistical analysis, it was determined that the dentists who participated in this study regarded the improper disposal of bio-hazardous waste as the primary factor contributing to the adverse environmental effects resulting from dental practices (87.1% of 70 possible affirmative answers). The participants in this study also identified energy, electricity, water and paper waste as the second factor that had a detrimental impact on the environment, with a percentage of 67.1% of 70 possible affirmative answers. Our findings are in accordance with relevant studies found in the scientific dental literature [40,41,42,43]. Richardson et al. (2016) [42] highlighted that sustainable dental practices in the United Kingdom can be adopted by implementing environmentally inclined waste management strategies. In the same year, Grose et al. (2016) [43] pointed out that dental staff from the National Health Services (NHS) dental teams and private dental clinics in North Devon, United Kingdom, were aware of the concept of sustainability in dentistry and paid primary focus to the waste management protocol. Results from another study carried out in Iran showed that over half of the dental clinics involved in the study did not have any programs implemented for reducing or recycling waste [44].

According to the results of the present study, the participants perceived that the disposal methods employed for waste had a greater impact on sustainability than adopting a preventive approach to managing wastage of resources; however, the dental scientific literature emphasizes the importance of prioritizing waste reduction efforts over waste disposal in order to mitigate environmental impact [40]. It is more important to apply waste reduction strategies (such as reducing purchases, consumption, and improving stock management) and to adopt sustainable practices (such as utilizing durable equipment and reusable tools and instruments, reducing energy, electricity, water, and paper usage, and reducing medicine waste) to curb waste generation [40]. To optimize environmental and economic benefits, it is essential to prioritize the minimization of purchases and usage of items whenever possible [40]. This approach takes precedence over the challenge of managing large quantities of waste generated by dental practices. Based on the results of our study, it is apparent that a more comprehensive approach to pre- and post-graduate education is necessary to promote sustainability in dentistry. Such an approach should focus on enhancing the awareness and understanding of the key concepts of sustainability in dental practices. By implementing this education, future dental professionals can develop a deeper appreciation of the environmental impact of dental practices and become better equipped with the necessary knowledge and skills to implement sustainable practices in their workplaces.

Additionally, the improper use and recycling of amalgam for dental fillings was perceived by the participants in this study as an important factor within their dental activities that negatively impacts the environment. Amalgam used for dental fillings is a well-known health hazard due to its high mercury content; it must be separated from dental waste by means of appropriate separators [41,45,46]. It is acknowledged that the use of dental amalgam is currently very limited; various studies worldwide indicate that dental professionals consider that amalgam has a negative effect on the environment to a small or moderate extent [47,48,49]. On the other hand, the choice to use amalgam as a dental filling depend on “shared decision-making between dental providers and patients in the clinic setting, and local directives and protocols” [50].

Moreover, according to the participants of our study, the use of toxic products for chemical sterilization, conventional radiological systems and disposable items during dental procedures were considered less significant than the previously mentioned factors, although these practices may have an important contribution to environmental harm. On the other hand, our results indicated that older dentists participating in this study and those with a longer duration of professional activity possess a greater level of awareness about the negative environmental impact resulting from dental activity (*p* = 0.001, respectively *p* = 0.006). Moreover, the use of traditional radiological systems or consumable items during dental procedures were considered to have a negative impact on the environment by dentists with a significantly higher age (*p* < 0.001) and with a longer period of professional activity (*p* = 0.003) than those who did not share this perspective. Our results indicated that there is a statistically significant positive correlation between the age of dentists and the duration of their period of professional activity (*p* < 0.001).

Furthermore, the statistical score calculated to quantify the overall level of awareness regarding the negative impact of dental activity on the environment (SCORE_IM) indicated a moderate level of awareness. This result is in accordance with the statistical score that indicated a moderate level of awareness regarding sustainability among participants (SCORE_ED).

However, it is noteworthy that our findings reveal a considerable level of interest in sustainable dentistry in private practices, with 61.4% of the participants indicating such concerns. This result is particularly relevant considering that 58.6% of the subjects demonstrated an insufficient level of awareness concerning the concept of sustainability, and the constructed statistical scores indicate a moderate level of awareness regarding sustainability among participants. Nevertheless, these findings can be interpreted as being uplifting, for even though a considerable amount of people are not well aware of what sustainable dentistry consists of, they reported concerns for sustainable dentistry at their workplace and displayed eagerness towards gaining a better understanding of the concept. These outcomes indicate the necessity for education and training for sustainability, as highlighted in previous scientific studies [10,12,20,28,51,52], by implementing actions directly linked to sustainability: educational projects, workshops, theoretical courses, elective courses, interactive seminars, proper modification of the university curricula, etc.

### 4.3. Dentists’ Opinion Regarding the Most Effective Approaches for Integrating Sustainability into Dental Practices

The predominant responses of participants concerning the most effective approaches for implementing sustainability into dental practices were as follows: the reduction of paper usage (51 selected answers—72.9% of 70 possible answers), the reduction of energy/electricity consumption (44 selected answers/62.9% of 70 possible answers), and the utilization of modern digital equipment and technologies (42 selected answers/60% of 70 possible answers). Nonetheless, these above-mentioned viewpoints expressed by the participants align with the scientifically proven most effective strategies for fostering sustainability in the dental field, as documented in specialized literature [10,12,14,41,53]. The considerable amount of participants interested in the reduction of paper usage potentially indicates the need to accelerate the adoption and development of digital alternatives in Romanian dental practices, as dentists are overloaded with paper documents that require manual completion and archiving. The implementation of computer-based record systems could reduce the amount of time needed to update patients’ records, while also displaying changes [43].

The ranking of the “reduction of the use of disposable items during dental procedures” as the least preferred strategy for sustainability integration in dental practices, as reported by participating practitioners, is a thought-provoking and noteworthy finding since the strategies of “reduce, reuse, recycle and rethink” (commonly grouped together or considered as separate entities) are acknowledged as essential for obtaining a sustainable dental activity [14,54].

The analysis of responses from participants pertaining to their intended actions for implementing sustainability in their dental practices revealed a generally even and consistent distribution of opinions. Each option was selected with a comparable frequency to the others, as follows: installation of dental equipment with high energy efficiency—45 selected answers (64.3% of 70 possible answers); implementation of practices related to reusing and recycling—39 selected answers (55.9% of 70 possible answers); adoption of strategies focused on energy and water resource conservation—37 selected answers (52.9% of 70 possible answers); and the use of environmentally friendly or “green” products—37 selected answers (52.9% of 70 possible answers). The excessive use of personal protective equipment (PPE) was ranked lowest in the hierarchy of responses to the question (14 selected answers/44.3% of 70 possible answers); nevertheless, the selection of this option by some participants indicates their complete lack of comprehension of sustainability.

On the other hand, dentists who selected the adoption of strategies focused on energy and water resource conservation as an approach to protect the environment had a significantly higher age (*p* = 0.048) and longer period of activity (*p* = 0.023) in comparison to those who did not share this viewpoint. As previously mentioned, the older dentists and those with more extensive professional experience were more aware of the environmental impacts of dental procedures compared to the younger dentists. When compared to younger dentists, the older dentists’ options demonstrated a certain level of coherence, consistency and homogeneity, indicating a deeper awareness of sustainability within the dental domain. Older dentists’ larger educational background and significant professional expertise, which has been accumulated over many years, may be the source of their attitude towards sustainability.

Nonetheless, following our analysis, a few relevant elements can be discerned, such as the participants’ inclination towards the “installation of dental equipment with high energy efficiency” as the most prevalent option in the hierarchy of responses, followed by the “reduce-reuse-recycle” system and by the use of environmentally sustainable or “green” products. The potential challenges associated with adoption of high energy-efficient dental equipment are related to substantial long-term economic and environmental benefits [1,12,14]. It is acknowledged that the use of various equipment for diagnosis and treatment is the core of the dental practice and high energy intensive, therefore the use of highly energy-efficient dental equipment is very important; the following should be taken into consideration: higher upfront costs, limited availability on some markets, training requirements, potential financial incentives on some markets, etc. [1,12,14].

Our results also revealed that educating dental employees and patients about sustainable practices was not thought to be a successful method for integrating sustainability into dental offices (the lowest score reported for this strategy confirms the fact). There exists a lack of understanding and knowledge on how to become more environmentally sustainable in many fields of human activities [47,55,56]. Awareness through education is the essential key in all domains, and this should be the foundation of future strategies, as other authors have also mentioned [14,47]. On the other hand, a study conducted by Grose et al. (2016) [18] in England, UK indicated that the participants exhibited a positive disposition towards the notion of sustainability in dentistry; nonetheless, the analysis revealed that the participants also expressed criticism concerning the absence of explicit guidelines for sustainable practices within the field of dentistry. As it is mentioned in other relatively recent studies [28,57], there is interest among dental professionals in achieving environmental sustainability, however, there is insufficient awareness on how to best accomplish this.

To our understanding, sustainability in dentistry is multifaceted and involves many inter-connected stakeholders, including national governments, scientists, educators, manufacturers, distributors, dental equipment technicians, waste collectors and processors. For example, the governments have a significant role by promoting and regulating dental practice activities. The dental product manufacturers and suppliers can contribute by producing in a sustainable manner and distributing dental materials and equipment in an eco-friendlier way. Similarly, university teachers and scientists play an important role in the development of new, sustainable dental materials and equipment, and promoting dental sustainability among students and residents.

### 4.4. Dentists’ Opinion the COVID-19 Pandemic’s Impact on Sustainability in Dentistry

The results of this study indicated that the participants considered that the COVID-19 pandemic had a medium–high impact in terms of sustainability in the field of dental activity. In line with other scientific studies [13,15,58,59], the current analysis reveals that the COVID-19 pandemic had a medium influence on sustainability in dentistry according to 51.4% of participants, while 45.7% of them considered that the pandemic had a high influence on it. In order to avoid aggravating the pandemic’s epidemiology, dental practitioners needed to make intense preventive care efforts [60,61]. The COVID-19 pandemic had a negative impact on sustainability in dentistry, health professionals considerably increasing their usage of PPE materials, as a result of the pandemic, in order to lower the risk of infections [62]. On the other hand, it was demonstrated that dental staff, academic personnel, dental students and dental researchers were affected by the COVID-19 pandemic on multiple levels (such as economic, financial, social and emotional status) [63,64,65,66,67,68].

### 4.5. Limitations of the Study and Future Perspectives

Elaborating this pilot study allows us to learn and prepare for further studies. Our pilot study was conducted over a brief period of time and involved a reduced number of participants; it may be expanded over time and could be conducted on a representative sample of dentists, in other cities across Romania or even abroad, potentially facilitating the groundwork for comparative research. In addition, it is our intention to investigate more issues related to sustainability in dentistry (for example, transportation and frequency of visits to the dental practice) and to outline the various obstacles and challenges associated with the implementation of sustainable dental practice in a forthcoming study.

Furthermore, our study highlights the necessity for specialized education and training in sustainability in medical schools and professional organizations, ensuring that dental students and practitioners have both professional expertise and fundamental aptitudes in varied forms of sustainable entrepreneurship.

## 5. Conclusions

Following the evaluation of the obtained results, and given the limitations of the present study, we have reached the following conclusions:The participants demonstrated a moderate level of awareness regarding the concept of sustainability in dentistry;Older participants with a more extended professional activity were significantly more aware of the concept of sustainability in dentistry and of the environmental impact of dental activity when compared to the younger participants;Improper biohazardous waste disposal was identified as the primary factor contributing to the negative environmental effects resulting from dental practices; participants considered the installation of dental equipment with high energy-efficiency as the most important action to implement sustainability in their dental practices;Our study highlights the need to increase dental professional awareness to be environmentally sustainable, to implement more sustainable practices and to educate oral health professionals in the field of sustainability. Through the understanding of dentists’ attitudes toward sustainability in dentistry, good practices for educating current and future practitioners on environmental sustainability can be formulated.

## Figures and Tables

**Figure 1 healthcare-11-01326-f001:**
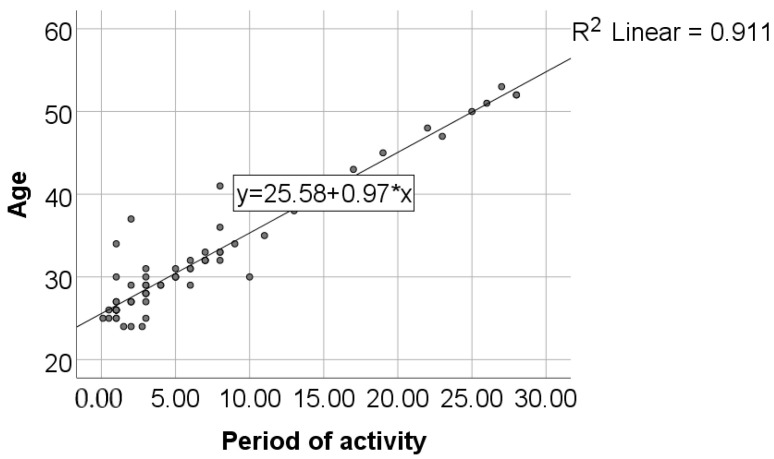
Correlation between dentists’ age and the duration of their period of activity.

**Figure 2 healthcare-11-01326-f002:**
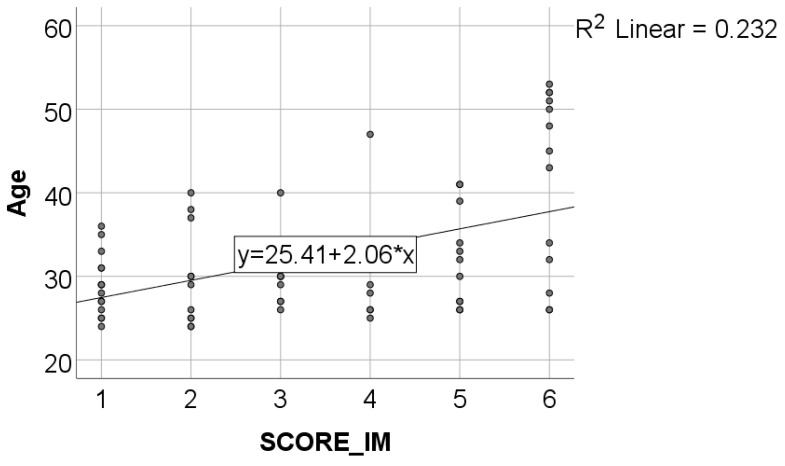
Correlation between dentists’ age and SCORE_IM.

**Figure 3 healthcare-11-01326-f003:**
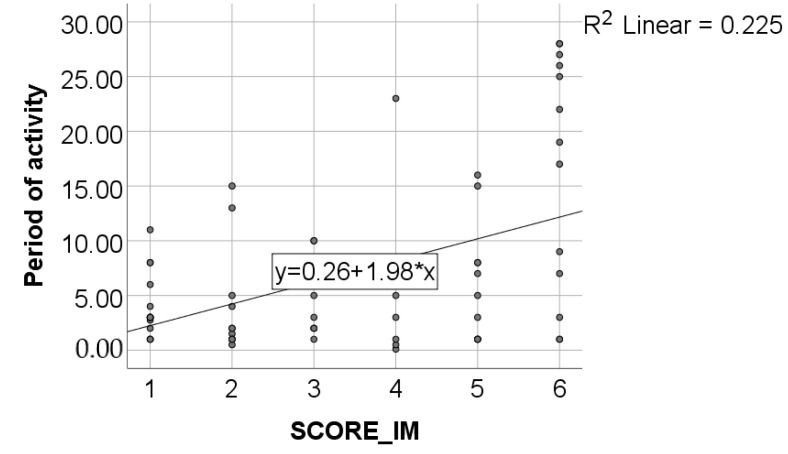
Correlation between the duration of dentists’ period of activity and SCORE_IM.

**Figure 4 healthcare-11-01326-f004:**
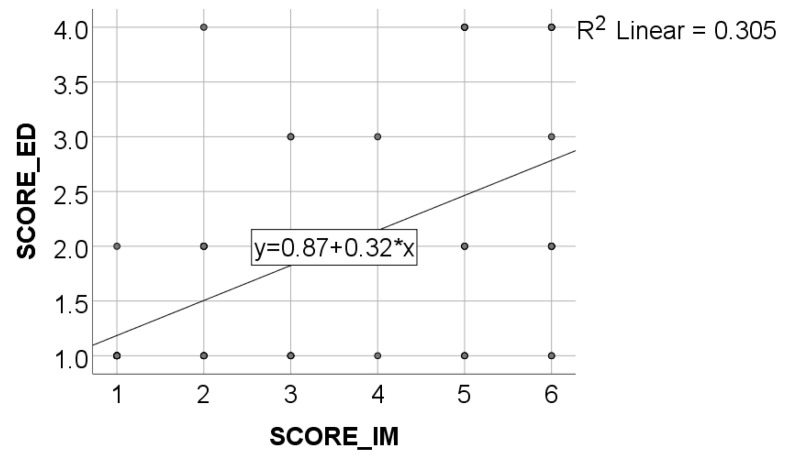
Correlation between SCORE_ED and SCORE_IM.

**Figure 5 healthcare-11-01326-f005:**
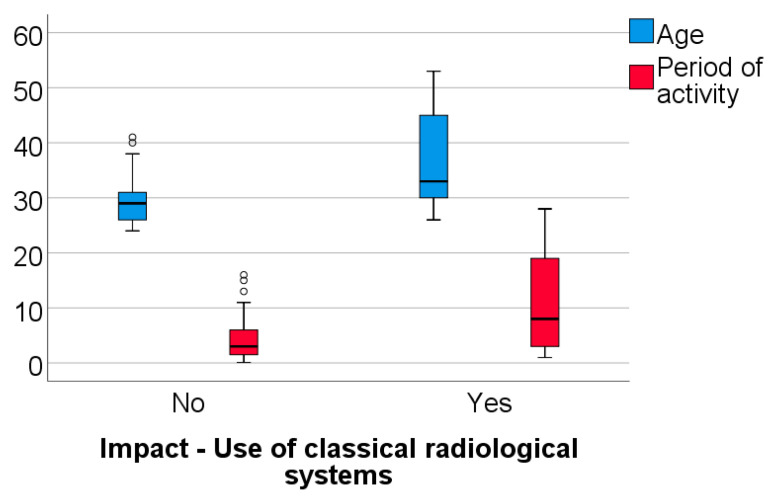
Comparison between the dentists’ age, the duration of their professional activity and answers related to the negative impact of the use of traditional/conventional radiological systems on the environment.

**Figure 6 healthcare-11-01326-f006:**
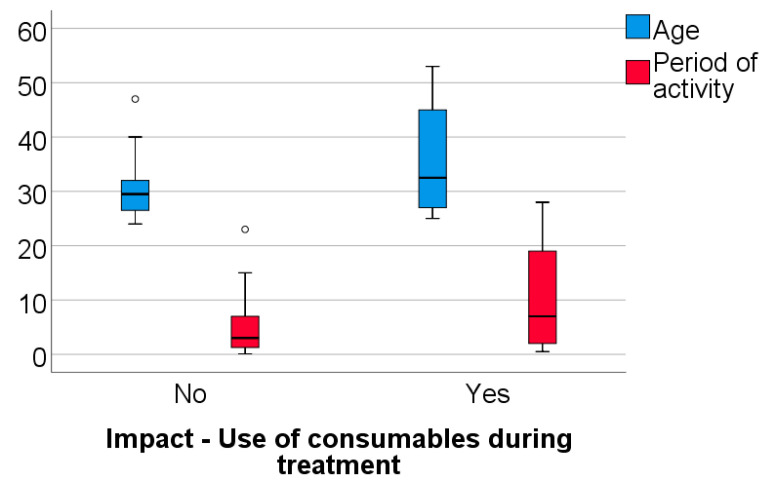
Comparison between the dentists’ age, the duration of their professional activity and answers related to the negative impact of the use of disposables on the environment.

**Figure 7 healthcare-11-01326-f007:**
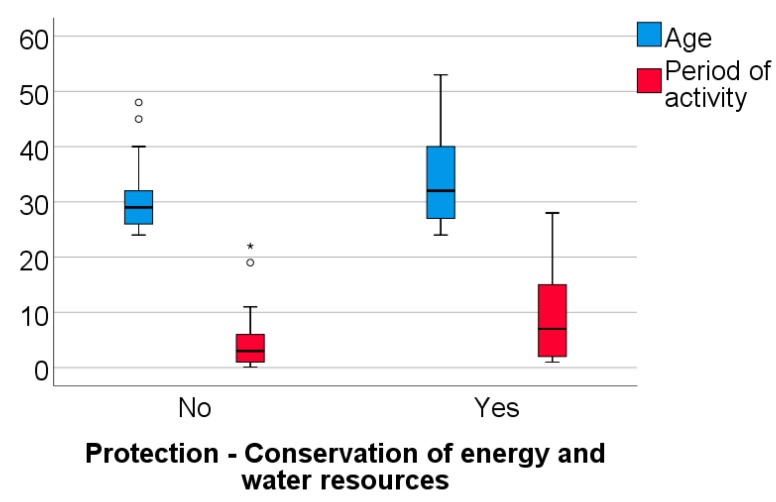
Comparison between the dentists’ age, the duration of their professional activity and answers related to energy and water conservation as an approach to protect the environment.

**Figure 8 healthcare-11-01326-f008:**
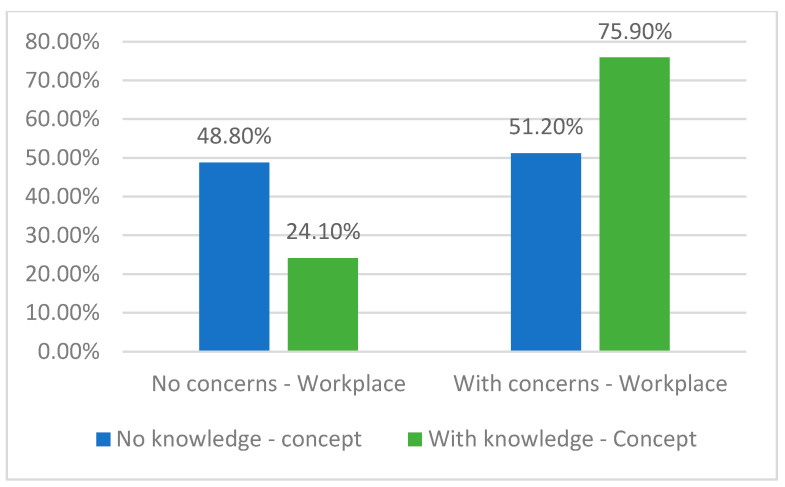
Distribution of the dentists according to their awareness of the concept of sustainability in dentistry and the concerns for sustainability in their workplace.

**Table 1 healthcare-11-01326-t001:** Distribution of the dentists’ answers from the applied survey.

Investigated Aspects	Questions (Q)	
**(1) Sociodemographic data**	**Q1. Please enter your age** **Q2. Please enter your gender** **Q3. Please specify the duration of your professional activity in years**	
**Investigated** **aspects**	**Questions (Q) and possible answers**	**Affirmative answer (No., %)**
**(2) Awareness on sustainability**	**Q4: Are you well aware of the concept of sustainability in dentistry?**	29 (41.4%)
	**Q5: What are the aspects which represent sustainability in today’s dental practice?**	
	a. Reduction of the impact of dental activity on the environment	58 (82.9%)
	b. Reduction of the amount of wasted energy, electricity, water and paper	44 (62.9%)
	c. Use of digital equipment, technologies and workflows	21 (30%)
	d. Use of modern dental materials	14 (20%)
**(3)** **Negative impact of dental activities on environment**	**Q6: What are the most important factors related to your dental activity which negatively impact the environment?**	
	a. Improper bio-hazardous waste disposal	61 (87.1%)
	b. Energy, electricity, water and paper waste	47 (67.1%)
	c. Improper use and recycling of dental amalgam	41 (58.6%)
	d. Chemical sterilization with toxic products	35 (50%)
	e. Use of traditional/conventional radiological systems	29 (41.4%)
	f. Use of disposable items during dental procedures	26 (37.1%)
	**Q7: Are there any concerns for the implementation of sustainable dentistry in your workplace?**	43 (61.4%)
**(4) approaches for integrating sustainability into dental practices**	**Q8: What are the most effective approaches for implementing sustainability into dental practices? (select ≤ 3 answers)**	
	a. Reduction of paper usage	51 (72.9%)
	b. Reduction of energy/electricity consumption through the use of modern lighting equipment	44 (62.9%)
	c. Use of digital equipment and technologies (digital radiographs, intraoral scanners, CAD/CAM technology, 3D-printers etc)	42 (60%)
	d. Reduction/phasing down of the use of dental amalgam and improvement of its recycling	28 (40%)
	e. Reduction of water consumption	20 (28.6%)
	f. Reduction of the use of disposable items during dental procedures	16 (22.9%)
	**Q9: What actions do you intend to take so as to implement sustainability in your dental practice? (select ≤ 3 answers)**	
	a. Installation of dental equipment with high energy efficiency	45 (64.3%)
	b. Implementation of practices related to reusing and recycling	39 (55.7%)
	c. Adoption of strategies focused on energy and water conservation	37 (52.9%)
	d. Increase in the use of environmentally friendly or “green” products	37 (52.9%)
	e. Sustainable education of both dental staff and patients	31(44.3%)
	f. Excessive use of PPE	14 (20%)
**(5) the COVID-19 pandemic’s impact**	**Q10. How do you assess the impact of the COVID-19 pandemic in your field of activity in terms of sustainability?**	
	Low impact	2 (2.9%)
	Medium impact	36 (51.4%)
	High impact	32 (45,7%)

**Table 2 healthcare-11-01326-t002:** Correlations between analyzed parameters.

Correlation	*p* *
Age x Period of activity	<0.001, R = 0.879
Age x SCORE_IM	0.001, R = 0.392
Period of activity x SCORE_IM	0.006, R = 0.328
SCORE_ED x SCORE_IM	<0.001, R = 0.571

* Spearman’s rho correlation coefficient.

**Table 3 healthcare-11-01326-t003:** Comparison between dentists’ age, their period of activity and certain answers in the survey.

Question/Age, Period of Activity (Median, IQR)		Age	Activity
Impact—Use of classical radiological systems	No	29 (26–31)	3 (1.25–6)
	Yes	33 (29–46)	8 (2–20.5)
	*p* *	<0.001	0.003
Impact—Use of disposables during dental treatments	No	29.5 (26.2–32)	3 (1.12–7)
	Yes	32.5(26.7–45.7)	7 (1.75–19.75)
	*p* *	0.038	0.039
Adoption of strategies focused on energy and water conservation	No	29 (26–32)	3 (1–6)
	Yes	32 (27–40.5)	7 (2–15)
	*p* *	0.048	0.023

* Mann–Whitney U Test.

**Table 4 healthcare-11-01326-t004:** Distribution of the dentists according to certain answers in the survey.

Answers (No., %)		Are You Well Aware of the Concept of Sustainability in Dentistry?		*p* *
		No	Yes	
Are there any concerns for implementation of sustainable dentistry in your workplace?	No	20 (48.8%)	7 (24.1%)	0.037
	Yes	21 (51.2%)	22 (75.9%)	

* Pearson chi-square test.

## Data Availability

The data presented in this study are available on request from the corresponding authors.

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
