# Peer review of "Aspects Regarding Sustainability among Private Dental Practitioners from Bucharest, Romania: A Pilot Study"

_healthcare, 2023, doi:10.3390/healthcare11091326_

Round 1

Reviewer 1 Report

This paper presents the findings of a pilot survey of a selected group of general dental practitioners based in Bucharest on the concept of sustainability. Using a standardised self-completed electronic questionnaire, the authors conclude that less than half of the respondents were well aware of sustainability and that the key step for reducing the environmental impact of their activities lies with the installation of high energy-efficient dental equipment.

While the paper is important there are a number of areas which the authors should consider altering prior to any recommendation for publication. These are:

The term sustainability has a wider concept in dental care than that suggested by the authors. What the authors have explored are the activities that occur within a dental practice and are heavily skewed towards treatment interventions. This misses aspects of sustainability covering such issues as transport and frequency of visits to the dental practice. Given the nature of the data collected this aspect cannot be addressed and as such, the paper needs to be reworded to recognise the narrow focus of the authors work.

The second aspect concerns the sample of respondents. While demographic aspects of the sample are provided, there is no placement of this subgroup in the wider population. What are the demographics of the dental practitioners in Bucharest? Do the characteristics match the sample?

The wording of the questions is at best vague. It would appear that the key question, (question 4), which underpins the central aim of the work asks the respondent whether they are ‘well aware’ of the concept of sustainability. It is not clear what ‘well aware’ as opposed to being ‘aware’ is and whether the difference is important. Again given the authors cannot change the data collected, they need to recognise this issue and comment accordingly. Indeed, given that the work is a pilot study the authors should comment on lessons learnt from the current process and make suggestions.

The statistical analyses are too detailed for the current work. I would suggest that a more descriptive set of analyses are undertaken and the section shortened.

Finally, the discussion section is far too long and in some ways inappropriate. While it needs to cover the findings and some of the issues arising it must address shortcomings in the methods used given that it is a pilot study  which ideally would provide lessons for the main study. 

There are a few spelling and grammatical errors but the paper needs work to reduce the overall emphasis

Reviewer 2 Report

The presented research article reports the pilot study on the knowledge of sustainability among Dental Practitioners from Bucharest, Romania. The research was conducted in a very well-mannered, defined, and presented appropriately throughout the manuscript. However, the present study results were not significant to be considered for this journal. As mentioned by the authors most of the results were found to be similar to previous reports. Moreover, the subject's recruitment criteria were restricted and a small number of the population was used. The present study results can be enhanced by combining the follow-up study results, i.e., after taking the appropriate measurements to overcome the present study (already known) results, which would substantially improve the "knowledge of sustainability" of dental healthcare professionals.

The specific comments see attachment.

Round 2

Reviewer 1 Report

This paper is a resubmission of an earlier version. The authors have undertaken a number of changes which were identified in a previous report that in consequence has improved the both the readability and the content. However, there are a number o issues which remain that the authors should consider.

Perhaps most importantly, the emphasis of the current paper is on the findings from the study which is a pilot study. This should form the main discussion of their work with implications for the required substantive study to establish the beliefs of dentists in Romania. This also address the second issue in that the results are of dentists from Bucharest. The findings from the present study may or may not apply to the wider dental profession in Romania but without including representatives in their sample this cannot be established. As such the conclusions should discuss the implications for a subsequent study to establish the current understanding of the profession not the precise details of the data on beliefs. 

The authors should also emphasis that the study explores the sustainability of actions within a dental practice not the wider issues. While they have recognised this in parts of the current paper, the outstanding question is whether when undertaking the study to establish the views, they will, explore the wider issues or not. If they do not wish to include transport etc., then the authors need to modify the title of their work to recognise the limited coverage of sustainability. Currently this is not the case. 

There are a few aspects that require the english modifying. Overall the standard for non-english based research is very good.

Reviewer 2 Report

Thank you for your answers. I could see that the manuscript has been considerably improved. 

Author Response

Dear Reviewer,

Thank you again for your effort and contribution to the improvement of our paper.

The Authors